# Single-Cell Transcriptomics of *Mtb*/HIV Co-Infection

**DOI:** 10.3390/cells12182295

**Published:** 2023-09-17

**Authors:** Smita Kulkarni, Janice J. Endsley, Zhao Lai, Todd Bradley, Riti Sharan

**Affiliations:** 1Texas Biomedical Research Institute, San Antonio, TX 78227, USA; 2Departments of Microbiology & Immunology and Pathology, The University of Texas Medical Branch, Galveston, TX 77555, USA; jjendsle@utmb.edu; 3Greehey Children’s Cancer Research Institute, The University of Texas Health San Antonio, San Antonio, TX 78229, USA; laiz@uthscsa.edu; 4Genomic Medicine Center, Children’s Mercy Research Institute, Children’s Mercy Kansas City, Kansas City, MO 64108, USA; tcbradley@cmh.edu; 5Departments of Pediatrics and Pathology and Laboratory Medicine, University of Kansas Medical Center, Kansas City, MO 66160, USA; 6Department of Pediatrics, UMKC School of Medicine, Kansas City, MO 64108, USA

**Keywords:** TB/HIV co-infection, single cell analysis, latent TB infection, biomarkers

## Abstract

Tuberculosis (TB) and Human Immunodeficiency Virus (HIV) co-infection continues to pose a significant healthcare burden. HIV co-infection during TB predisposes the host to the reactivation of latent TB infection (LTBI), worsening disease conditions and mortality. There is a lack of biomarkers of LTBI reactivation and/or immune-related transcriptional signatures to distinguish active TB from LTBI and predict TB reactivation upon HIV co-infection. Characterizing individual cells using next-generation sequencing-based technologies has facilitated novel biological discoveries about infectious diseases, including TB and HIV pathogenesis. Compared to the more conventional sequencing techniques that provide a bulk assessment, single-cell RNA sequencing (scRNA-seq) can reveal complex and new cell types and identify more high-resolution cellular heterogeneity. This review will summarize the progress made in defining the immune atlas of TB and HIV infections using scRNA-seq, including host-pathogen interactions, heterogeneity in HIV pathogenesis, and the animal models employed to model disease. This review will also address the tools needed to bridge the gap between disease outcomes in single infection vs. co-infection. Finally, it will elaborate on the translational benefits of single-cell sequencing in TB/HIV diagnosis in humans.

## 1. Introduction

The Tuberculosis (TB) and Human Immunodeficiency Virus (HIV) co-pandemic continues to pose a major healthcare burden in resource-limited countries [1]. TB is caused by *Mycobacterium tuberculosis* (*Mtb*), which spreads from person to person through aerosol droplets when the infected person coughs or speaks [2]. According to the Global Tuberculosis Report by the World Health Organization (WHO), an estimated 187,000 people with HIV died of TB in 2021 [3]. People living with HIV (PLHIV) are more likely to develop active TB disease than those without HIV [4,5]. This is due to several factors, including increased susceptibility to new *Mtb* infection and, especially, the reactivation of the untreated latent TB infection upon immunosuppression by HIV co-infection [6]. 

Both pathogens attack the immune subsets needed for protection in the human body: macrophages and CD4^+^ T cells [7,8]. However, the mechanisms of this manipulation of the immune responses are not well understood. HIV-1 depletes CD4+ T cells and infects macrophages, leading to virus persistence in reservoirs. *Mtb* pathogenesis is characterized by phagocytosis of the inhaled bacilli by alveolar macrophages. Upon internalization, *Mtb* is either cleared, or it proliferates within the macrophages [9]. One of the hallmark features of the immune response to *Mtb* is the formation of granuloma consisting of macrophages, multinucleated giant cells, Foamy cells, and lymphocytes [10]. Though the primary function of granuloma is to contain the bacilli, in some instances, the bacilli can survive inside these structures for extended periods of time in a dormant state. Under immunosuppressive conditions, such as HIV co-infection, the bacilli escape the granuloma and spread to extrapulmonary organs causing the reactivation of LTBI [6,11,12]. Nonhuman primate (NHP) research has shown that there are direct cytopathic effects of Simian Immunodeficiency Virus (SIV) resulting in chronic immune activation, altered effector T cell phenotypes, and dysregulated T cell homeostasis that causes LTBI reactivation [13]. Studies have also shown that while HIV-1 can replicate in the tuberculous microenvironment [14], *Mtb* is able to exacerbate HIV-1 pathogenesis through the formation of membranous structures in human macrophages [15]. Concomitant to this, PLHIV with LTBI have been shown to have elevated levels of immune activation and inflammation that impacts the progression of both diseases [16,17]. 

Highly effective combinatorial antiretroviral therapy (cART), while effective in reducing viral loads in the periphery and lungs of *Mtb*/SIV co-infected macaques, fails to reduce the rate of reactivation of LTBI [18,19]. Despite being on cART, people living with HIV (PLHIV) have increased risk of LTBI reactivation. This is partly due to the inability of cART to significantly reduce the virus-driven chronic immune activation that persists despite undetectable HIV viral loads [20,21]. This is, in turn, due to a loss of mucosal integrity in the gastrointestinal tract following HIV infection. This leads to the release of microbial products into the circulation, in turn leading to the systemic activation of a wide array of immune cells, such as T, B, NK, plasmacytoid dendritic cells (pDCs), and monocytes [1,22]. Additionally, the initiation of cART in humans with chronic HIV-1 infection can lead to a paradoxical worsening of TB and manifest as an immune-reconstituted inflammatory syndrome (TB-IRIS) [23]. 

The characterization of individual cells using next-generation sequencing-based technologies has allowed for previously unknown biological discoveries. RNAseq of bulk cellular populations has been successfully utilized in the past to perform population-level gene expression studies. In comparison to the more conventional sequencing techniques that provide a cellular population-level assessment, single-cell RNA sequencing (scRNA-seq) has the advantage of revealing complex and new cell types as well as identifying cell-to-cell variability. scRNA-seq provides significant resolution for answering the key biological questions involving cell heterogeneity and profiling differences in cellular communities [24,25]. Meaningful cell-to-cell gene expression variability has been revealed in recent years by applying scRNA-seq to diverse human illnesses, including TB and HIV [26,27,28,29,30]. Clearly, scRNA-seq presents a novel and unprecedented opportunity to study the genome, transcriptome, and other single-cell multi-omics. In this review, we will summarize the application of scRNA-seq to reveal host–pathogen interactions in *Mtb* and HIV infection and investigate the heterogenous cell populations in TB and HIV. Additionally, we will outline the animal models utilizing scRNA-seq to identify target biomarkers of human disease, including diagnostic and therapeutic markers. Finally, we will summarize the gaps in knowledge that need to be addressed in future studies. 

### Limitations and Challenges of LTBI Screening in HIV Patients 

A major challenge in the field is the lack of biomarkers to distinguish active TB from LTBI and predict reactivation upon HIV co-infection. LTBI management requires national LTBI policies and a standardized monitoring system to be in place in all countries [31]. However, effective LTBI screening and treatment remains a challenge in highly endemic nations. Additionally, high-income countries with low TB/HIV burden report the unsatisfactory implementation of LTBI screening in HIV clinics [32,33,34]. The centers for disease control and prevention (CDC) recommend that TB household contacts, people from highly endemic nations, people who work in high-risk settings such as homeless shelters, health care workers who care for patients at high risk of TB, and infants or children who are exposed to adults at high risk for LTBI, should be screened for LTBI. They also recommend that all people with HIV should be evaluated for LTBI at the time of HIV diagnosis [35,36]. The two types of available TB screening tests are: the TB skin test and TB blood test (interferon-gamma release assays or IGRAs); and the QuantiFERON^®^-TB Gold Plus (QFT-Plus, Qiagen, Germantown, MD, USA) and the T-SPOT^®^.TB test (T-Spot, Oxford Immunotec Ltd., Abingdon, UK). Both TST and IGRAs can only detect *Mtb* infection and fail to differentiate LTBI, LTBI reactivation, and/or TB progression [37]. TST measures the response to the purified protein derivative (PPD) as the form of a delayed-type hypersensitivity response [38]. However, TST lacks specificity, and produces false positives with Bacillus Calmette–Guérin (BCG) vaccination and false negatives in immunosuppression [39,40,41,42,43]. There have been studies on IFN-γ release assays in the diagnosis of LTBI in HIV-infected populations [44,45,46] but more studies are needed to analyze the impact of immunosuppression on the antigen-specific T-cell responses. Steffen et al. [47] analyzed more specific and cost-effective strategies for the diagnosis of LTBI in PLHIV in Brazil. The strategies included TST with the new PPD RT 23, two novel skin tests (Diaskintest, Generium Pharmaceutical, Moscow, Russia), EC (Zhifei Longcom Biologic Pharmacy Co., Hefei, Anhui, China), and the QuantiFERON-TB-Gold-Plus. They concluded that the newer tests were more sensitive and cost-effective for the Brazilian population compared to TST. Recombinant *Mtb* fusion protein (ESAT6/CFP10) is the latest method to detect LTBI. EC has shown consistency compared with IGRAs in detecting LTBI in HIV, irrespective of immunosuppression status [48]. The new tests are designed to measure *Mtb*-specific immune responses to help avoid false positives such as those seen in TST. Additionally, they are also affordable, easy to handle, and higher in specificity than TST. However, some of the potential disadvantages include the insufficient differential diagnosis of LTBI and the impact of antigen-based vaccines on the diagnostic value of these new tests. Increased attention needs to be focused on omics technologies to understand the differential expression of various antigens in LTBI and active TB infection in studies with large sample sizes and in the setting of HIV co-infection.

## 2. scRNA-Seq—A Tool to Study Host–Pathogen Interactions and Biosignatures in TB

Pulmonary TB, caused by infection with *Mtb*, is primarily identified as a lung disease wherein the bacilli are able to evade the host immune system in ~5–7% of individuals and cause extensive inflammation and pathology [49]. In 90% of the infected individuals, the infection is controlled by the host immune response, and the mycobacterial replication is contained within granulomatous structures [50]. This results in LTBI that can either last for the lifetime of the individual or it can reactivate in an immunosuppressed condition, such as HIV co-infection [51]. The interaction of the immune cells and complex intracellular bacterial pathogens, such as *Mtb*, produces multidimensional interactions. Transcriptional profiling at the single-cell level within the dynamic granulomatous structure provides an opportunity to understand the infection as a consolidated process between the host and pathogen (Figure 1). Indeed, heterogeneity is integral to the interactions between intracellular bacteria and host immune cells such as macrophages [52]. Additionally, there is evidence that variation in the host and pathogen transcriptional programs causes differential infection outcomes [53,54]. Further, each granuloma within an individual differs substantially in its composition, inflammatory profile, size, and bacterial ecology [10]. Cellular compositions and cell-to-cell signaling networks have been identified using high-throughput scRNA-seq within TB lung granulomas in the cynomolgus macaque model of TB [55]. Granulomas with persistent bacteria were characterized as having TH2 immunity-based signaling among mast, endothelial, fibroblast, and plasma cells. Granulomas demonstrating bacterial control appeared enriched in cytotoxic T cells signaling via TH1, TH17 immunity [55]. In a similar approach to studying heterogeneity in the cellular composition of leprosy granulomas, scRNA-seq was utilized to perform Uniform Manifold Approximation and Projection (UMAP) dimensionality reduction and cell clustering [56]. This was followed by differential expression analysis to find the cluster markers that defined signature genes. This ultimately resulted in the identification of 66 key antimicrobial genes expressed by T cells, macrophages, keratinocytes, and fibroblasts that have been reported to participate in the immune response to mycobacterial infections [56]. By using scRNA-seq in combination with other technologies, both these studies were able to demonstrate the participation of the non-traditional immune cells in a spatial model of mycobacterial infection. Future studies must further elucidate these observations to outline the mechanisms involved in the HIV-1-mediated disruption of granulomas in LTBI.

Despite advances in understanding the host–pathogen interactions in TB, the lack of biomarkers to distinguish active TB from LTBI remains a major challenge for TB control. Individual immune cell subsets have been delineated using scRNA-seq on peripheral blood mononuclear cells (PBMCs) from healthy controls (HC), LTBI, and TB patients using an unbiased, surface-marker-free approach [57]. Identifying the nature and function of these circulating cell subsets provided a useful framework for examining TB disease progression. It also identified new cell type markers in PBMC as well as an altered T-cell subsets distribution in TB. A CD3-CD7+GZMB+ subset was found to be significantly decreased in patients with TB, compared to HC and LTBI controls, suggesting that this could serve as a distinguishing marker between TB and LTBI [57]. In another study, Xu et al. combined expression profiling by array and scRNA-seq to profile three key immune-related hub genes: ADM, IFIT3, and SERPING1, with the potential to discriminate TB from LTBI and HC [26]. In addition to identifying potential biomarkers to predict TB, NGS has also contributed to the characterization of certain circulating cell subsets involved in immune dysregulation in HIV-1 and TB co-infection [9]. Comparison of the transcriptome of PBMCs from patients with HIV-1 infection alone and those with HIV-1-TB co-infection revealed a higher proportion of inflammatory CD14+CD16+ monocyte subset in HIV/TB cohort. Thus, peripheral monocyte subsets might serve as a discriminating biomarker for the diagnosis of HIV/TB co-infection. Additionally, a biosignature comprising *RAB20* and *INSL3* genes in the peripheral blood was identified using scRNA-seq technology to accurately classify TB status among advanced HIV patients in two geographically distinct cohorts [58]. Yet, another study successfully identified that intermediate and classical monocytes contribute to the blood immune signatures of ATB in humans using scRNA-seq profiling [59].

Though scRNA-seq has proven to be instrumental in addressing cell-to-cell heterogeneity, it is imperative to consider this variability as a paradoxical phenomenon. The discovery of a pool of potential candidate biomarkers comes with a deluge of false positives. Interpreting the candidate genes requires careful functional considerations that can add another layer of complex statistical analyses. It is also important to consider that TB manifests itself at the proteome and whole organ levels, not just the transcriptome level. Hence, future NGS studies must aim to consolidate the biomarkers from the periphery and local [60] immune responses to present a comprehensive biosignature of TB progression in the setting of HIV co-infection. 

### An Insight into the Mechanisms of HIV Latency Using scRNA-Seq and the Effect of Co-Infections on HIV-Reservoir

Latent HIV proviruses persist long-term in a small population of memory CD4+ T cells called reservoirs. The reservoir cells present a major hurdle to eradicating HIV [61,62,63,64]. The latent reservoir is maintained long-term, enabling rapid virus rebound upon ART interruption [65,66,67,68,69,70]. Due to the lack of a single or a set of markers defining the HIV-harboring reservoir cells, their characterization is challenging. Various techniques were employed to enrich and sort latent virus-harboring cells in combination with scRNA-seq to advance our understanding of HIV persistence and reactivation upon co-infection.

HIV latency and its mechanisms were studied in a primary cell model of HIV by Bradley et al. [71]. The study revealed different viral and host gene transcription patterns within the latently infected cell subpopulations, which were identifiable by specific genes. Further, the HIV downregulation in primary cells was regulated by the underlying transcriptional program of the infected cells. Golumbeanu et al. used a primary CD4+ T-cell model of HIV latency and scRNA-seq and identified higher cellular activation, as indicated by global transcriptomic and metabolic activity in HIV-reactivated cells. This suggested that the cellular environment contributed to HIV reactivation [28]. 

Data from multiple studies indicate a clonal expansion of CD4+ T cells harboring intact or defective latent proviruses over time [61,62,72,73,74,75,76]. Several studies have also observed that the reservoir CD4+ T cell clones express the same unique T-cell receptor (TCR) and have a single, distinctive proviral integration site [68,72,74,77,78]. 

Liu et al. developed another method, Sortseq, which used HIV-RNA staining to identify and sort HIV-reactivated cells using Fluorescence-Activated Cell Sorting (FACS) [79]. This study demonstrated that HIV+ Sortseq cells enriched the T_H1_ phenotype, up-regulated cellular factors supporting HIV-1 transcription (*IMPDH1* and *JAK1*), or promoted cellular survival (*IL2* and *IKBKB*). Collora et al. used single-cell expanded CRISPR-compatible cellular indexing of transcriptomes and epitopes by sequencing (ECCITE-seq). This technique captured surface protein expression, cellular transcriptome, HIV-1 RNA, and TCR sequence within the same single cell in longitudinally archived paired samples during acute viremia and after viral suppression post-one-year ART [80]. They showed that post-suppressive ART, HIV resides in heterogenous Granzyme B+ T_H1_ effector memory CD4+ T cells with robust antigen responses, proliferation potential, and long-term clonal stability [80].

Intact proviruses are enriched among CD4+ CD45RA-HLA-DR+ memory T cells [65,81,82,83,84]. Weymar et al. enriched quiescent reservoir memory CD4+ T cells using their unique TCR as a molecular identifier. They showed that HIV-1 proviral integration and latency did not induce a specific transcriptional program [85]. They postulated some possibilities to explain why latent proviruses may be enriched in T_EM_ cells, such as that the T_EM_ cells may be among the most likely responders to chronic infection and become infected in the process; and the T_EM_ transcriptome may favor the suppression of HIV-1 gene expression during T-cell activation. Thus, the cells undergo clonal expansion in response to antigen stimulation without HIV virion production and cell death. In addition, T_EM_ cells may evade natural killer cytotoxicity contributing to their long-term survival. 

T-cell activation stimulates pathways that reactivate HIV from latency. Additionally, non-productive HIV-1 infection can also lead to bystander CD4+ T cell death by apoptosis or pyroptosis [86]. However, some latent cells divide and survive. The scRNA-seq analyses showed that reactivated latent cells could express a distinct transcriptional program that can suppress HIV-1 transcription, which may help their survival. Cohn et al. developed a method to capture reactivated latent cells by combining antibody staining of surface HIV env protein, magnetic enrichment, and flow cytometry (latent cell capture; LURE) [78]. Although LURE captured a fraction of circulating latent cells with proviruses that can be reactivated in a single round of stimulation, this study concluded that the transcriptome of peripheral blood latent T cells allows cell division without activating HIV-replication triggered cell death pathways. Recently, Clark et al. developed a microfluidic technique named Focused Interrogation of cells by Nucleic acid Detection and Sequencing (FIND-seq) [87] to isolate reservoir cells from HIV patients treated with ART long-term and to analyze their transcriptome at the single-cell level [88]. FIND-seq allowed them to study the transcriptome from cells containing quiescent virus without latency reversal. The transcriptome of HIV-DNA+ memory CD4+ T cells revealed inhibited pathways such as death receptor, necroptosis, and antiproliferative signaling but showed higher expression of the known negative regulators of HIV transcription [88]. Thus, the scRNA seq studies have shown that the transcriptome of latent HIV-harboring CD4+ T cells favors long-term cell survival, proliferation, and HIV-silencing and may provide targets for effective therapeutic approaches to clear the proviral infection. 

In addition to CD4+ T cells, monocytes expressing CD14+CD16+ are infected by HIV and contribute to establishing, reseeding, and maintaining viral reservoirs [89,90,91,92,93]. Leon-Rivera et al. used scRNA seq to detect HIV and host transcripts simultaneously in HIV-infected (HIV RNA+) and HIV-exposed monocytes from ART-treated or untreated patients [94]. This study highlighted that monocyte subsets provide varying degrees of permissive cellular environment for HIV infection, and ART has a detrimental effect on monocyte function [94]. Similarly, macrophages and microglia that constitute 5–10% of brain cells are the primary HIV-infected cells in the central nervous system (CNS). Thus, CNS is considered to be a large HIV reservoir. Even with the cART treatment, 20–50% of PLHIVs develop HIV-associated neurocognitive disorder [49]. Plaza-Jennings used single nucleus RNA-seq and found high levels of HIV transcription in activated microglia [95]. The results of this study indicate that HIV-induced interferon responses modulate host chromatin conformation and HIV integration sites in microglia. This study is limited by only mapping the nuclear RNA which, in the case of HIV, could mean that functional and defective RNA are retained in the nucleus. However, even with this caveat, this study expanded our understanding of HIV pathogenesis beyond peripheral blood lymphocytes. Overall, these studies demonstrated that when using scRNA-seq, there is significant heterogeneity in both active and latent HIV-infected cells. Understanding the phenotype of these cells at high resolutions could help better design tools for identifying these cell populations and developing therapies against *Mtb*/HIV co-infection. 

*Mtb* infection promotes HIV replication, associated with HIV viral load. *Mtb* infection leads to increased levels of IL-7 [96], which is shown to enhance the establishment and maintenance of the HIV-1 reservoir [97,98]. IL-17+ T helper cells (T_H17_) show increased susceptibility to HIV infection and further exploitation as a viral reservoir that persists despite cART [99]. Several studies indicate perturbations in IL-17 expression in *Mtb* infection [100,101,102]. *Mtb* also elevated indoleamine 2,3 -dioxygenase (IDO), which positively correlated with HIV persistence [103,104]. Population-level studies to determine the effect of *Mtb* co-infection on the HIV reservoir have reported contrasting observations. Host and viral genetics, pre-existing latent or active *Mtb* infection before HIV acquisition, and the initiation of cART could be some of the variables contributing to the effect of *Mtb* infection on HIV reservoirs, which future studies at the single-cell level could further illuminate. Over 70 percent of PLHIV live in Africa, a region where *Mtb* is endemic. Thus, determining the effect of *Mtb* on the HIV reservoir is essential to understand HIV persistence and develop strategies for HIV eradication in these populations. 

## 3. Predicting Disease Progression in TB and HIV Using scRNA-Seq

Infections are dynamic processes in which the outcomes are dictated by the complex interactions of the pathogen and the multiple biological factors of the host. scRNA-seq allows a deeper understanding of the human cellular landscape by studying cellular behavior at a higher resolution. The early diagnosis of LTBI and screening of those at a higher risk of HIV co-infection is imperative for TB prevention programs [105]. A simultaneous epitope and transcriptome measurement, using Cellular Indexing of Transcriptomes and Epitopes (CITE-seq), identified a T-cell state associated with TB progression that responded to ex vivo *Mtb* peptide stimulation [106]. The study was based on the fact that TB progression was largely influenced by host memory T cells, which had been shown previously. However, prior studies have failed to comprehensively describe the landscape of the progression-associated memory T cells in TB [107,108]. Most of the significant differences resided in a rare multimodally defined T_H17_ subset (C-12). The C-12’s abundance in patients that had recovered from TB indicated either a prior TB disease or susceptibility to TB disease progression. Additionally, mutations in the RAR-related orphan receptor C (RORC), that are highly expressed in C-12, were shown to increase susceptibility to TB [106,109]. The study provides a resource to enable further studies of the different memory T cell phenotypes that could serve as biomarkers for progression to TB in the setting of HIV co-infection. 

In a different approach by Ben-Moshe et al., scRNA-seq of human peripheral blood cells infected with *Salmonella* was used to develop a deconvoluted algorithm to study three cohorts of TB patients at different stages of the disease [110]. The trained algorithm predicted the power of the monocyte-infection-induced signature in classifying the LTBI individuals at a higher risk of developing active TB. The study highlights the role of global infection dynamics, such as the monocyte-infection-induced state in *Salmonella* and TB. The approach can be applied to additional co-morbid pathogens, such as HIV, and at additional time points in order to train the deconvolution further to differentiate the infection-induced states and infectious outcomes. Early interleukin-1 (IL-1) signaling enrichment using scRNA-seq has been implicated as an important pathway in response to *Mtb* stimulation in human peripheral blood cells [111]. The study provided a detailed analysis of the significance of cell-type specificity compared to pathogen-specificity and its role in influencing gene expression in TB. Concomitantly, elevated levels of monocytic IL-1b upon TLR-2 and TLR-4 stimulation have been shown to be associated with reduced odds of TB recurrence in ART-treated patients [112]. 

Innovative single-cell technologies indicate efforts to shift from traditional therapy in TB to precision medicine and host-directed therapeutics. scRNA-seq allows the simultaneous analysis of the host and pathogen transcriptomics on a wide array of samples ranging from primary cell cultures to animal tissue biopsies or blood from the human host. Future studies could focus on utilizing single-cell technology in combination with spatial transcriptomics to understand the impact of gut microbiota, especially in the context of HIV co-infection in humans. 

HIV acquisition and the clinical outcomes of HIV-1 infection are highly variable among individuals. Genetic factors, socio-economic backgrounds, and complex interactions between the virus and the host determine them. Besides the genetic variation of major host factors, such as human leukocyte antigen receptor (HLA) and HIV co-receptor CCR5, polymorphisms at several loci affect the outcomes of HIV infection as well as the pharmacokinetics of anti-HIV drugs (reviewed in reference [113]). Nevertheless, even among the most highly exposed individuals, a fraction remains HIV-negative without any known genetic association [114,115]. Similarly, important differences in the natural course of HIV infection (such as time from infection to AIDS diagnosis and opportunistic infections or malignancies) were only partially explained by known variables such as genetic polymorphism, age, and co-morbidities [116]. Single-cell transcriptome studies, performed in patient-derived cells and in vitro infected primary cells and cell lines, have significantly contributed to understanding the host and virus variability that may alter the overall clinical course of infection (Table 1; Figure 2).

CD4+ T cells are the main target of HIV infection. However, CD4+ T cells are not equally permissive to infection, varying between individuals and across cells isolated from the same individual [29,117,118,119]. Combining scRNA-seq and cell surface marker expression using flow cytometry, Rato et al. defined the ‘HIV-permissive cell’. Their data showed that the transcriptional heterogeneity at the single-cell level was contributed to mainly by the varying degrees of response to T-cell receptor (TCR) stimulation, allowing them to partition cell populations into high and low HIV-permissive subsets [120]. The scRNA-seq has been used to determine the cellular transcriptome changes induced by active HIV replication in macrophages [121], the multicellular immune responses during hyperacute HIV infection [30], the poor restoration of T cell function after ART in HIV-infected patients [122,123], immune exhaustion in chronic HIV infection [124], immune reconstitution failure in highly active antiretroviral therapy (HAART)-treated patients [125], and the collective effects of drug abuse and SIV infection in microglia and brain macrophages from SIV-infected rhesus monkeys [126]. All these contribute to our understanding of HIV progression by itself and make educated predictions of outcomes in the setting of *Mtb* co-infection. 

**Table 1 cells-12-02295-t001:** HIV acquisition and the clinical outcomes of HIV-1 infection.

Study	Technique	Observation	Ref.
Defining HIV-permissive cells	scRNA-seq	TCR stimulation mainly contributes to transcriptional heterogeneity in CD4+ T cells, which regulates HIV transcription.	[120]
Immune responses during hyperacute HIV infection	scRNA-seq on peripheral blood mononuclear cells from four untreated individuals before and longitudinally during acute infection	Immune cell responses to HIV infection within the first weeks of infection, such as proliferating natural killer cells, which potentially may be associated with viral control	[30]
Restoration of T cell function after ART	scRNA-seq on peripheral T cells on chronic HIV-infected treatment-naïve or ART-treated patients	Significant loss of naïve T cells, prolonged inflammation, and increased response to interferon-α in treatment-naïve patients, partially restored by ART. Granulosyn expressing CD4+ and CD8+ Effector cell clusters correlated with poor immune restoration	[123]
Immune Exhaustion in Chronic HIV Infection	scRNA-seq on peripheral blood mononuclear cells from HIV patients and healthy subjects	An inhibitory receptor KLRG1 was identified in an HIV-1 specific exhausted CD8+ T cell population expressing KLRG1, TIGIT, and Tbet^dim^ Eomes^hi^ markers	[124]
Immune Reconstitution failure in (HAART)-treated HIV patients	scRNA-seq and scATAC-seq analysis of peripheral blood mononuclear cells (PBMCs) derived from immune non-responder [127] and responder (IR)	Low expression of mucosal-associated invariant T (MAIT) cells in INRs, which exhibited transcriptional profiles associated with impaired mitochondrial function and apoptosis signaling	[125]
Cellular transcriptome changes induced by active HIV replication in macrophages	An in vitro system to model HIV-1 infection of macrophages and single-cell RNA sequencing (scRNA-seq) to compare the transcriptomes of uninfected cells, cells harboring pre-integration complexes (PIC), and those containing integrated provirus and making late HIV proteins	NFkB- and AP-1-promoted transcription characterize PIC cell transcriptomes, while E2F family transcription products distinguish transcriptomes of cells transcribing from provirus	[121]
Differential virus reactivation potential	Primary CD4+ T-cell model expression HIV green fluorescent protein (GFP); scRNA seq	Global transcriptomic profiles of cells with reactivated HIV showed higher cellular activation and metabolic activity	[28]
Characterize latent cells reactivated by a single round of stimulation	Latent cell capture; LURE. Purified CD4+ T cells from peripheral blood of ART-treated patients activated by LRS and sorted based on the expression of HIV-env protein and HIV-gag mRNA	CD4+ T cells harboring proviruses with identical Env sequences also showed identical TCRs. Reservoir cells arise by clonal expansion, and the reservoir is maintained by balanced cell division and cell death. Reactivated latent cells express a distinct transcriptional program that suppresses HIV-1 transcription, which helps them survive	[78]
Characterize reactivated cells within 24 h of latency reversal	Sortseq; stimulated peripheral blood CD4+ T cells from ART-treated, virally suppressed patients	HIV+ cells enriched in TH1 phenotype upregulate cellular factors that support HIV transcription and promote cellular survival. HIV promoter drives high aberrant host gene transcription downstream of the integration site	[79]
Transcriptome of quiescent reservoir memory CD4+ T cells harboring intact HIV provirus	Quiescent reservoir memory CD4+ T cells enriched using their unique TCR as a molecule, identified using scRNA-seq	HIV–proviral integration and latency did not induce a specific transcriptional program	[85]
Characterize HIV reservoir cells after suppressive ART-therapy	ECCITE-seq, which captures surface protein expression, cellular transcriptome, HIV-1 RNA, and TCR sequence within the same single cells in longitudinally archived paired samples during actual viremia and one-year post-ART	HIV revised in heterogenous granzyme B+ Th1 effector memory CD4+ T cells with robust antigen response, proliferation potential, and long-term clonal stability	[80]
Isolate reservoir cells from HIV patients treated with long-term ART	FIND-seq, simultaneous capture of polyadenylated RNA and HIV DNA from single cells, and scRNA-seq without latency reversal	HIV-DNA+ memory CD4+ T cells inhibited death receptor, necroptosis, and antiproliferative signaling but showed higher expression of known negative regulators of HIV transcription, thus favoring long-term survival of the cells and HIV silencing	[88]
The transcriptome of HIV-infected and HIV-exposed peripheral blood monocytes	scRNA seq	Monocytes provide varying degrees of permissive cellular environment for HIV infection, and ART has a detrimental effect on monocyte function	[94]
scRNA profiling of HIV-infected cells in CNS	Single nucleus RNA-seq of archived brain tissue	HIV-induced interferon response modulates host chromatin conformation and HIV integration sites in microglia	[95]

Wang et al. [124] investigated the role of HIV-1 infection on the role of immune cell exhaustion at the transcriptomic level in HIV disease progression using scRNA-seq. Three unique clusters of exhausted CD4+ and CD8+ T cells and interferon-responsive CD8+ T cells were identified in the PBMCs from HIV-infected donors compared to healthy donors. KLRG1+ expressing exhausted CD8+ T cell population was also identified as a critical disease progression marker that could serve to be an immunotherapy target for chronic HIV infection treatment. scRNA-seq has facilitated the identification of rare immune cell subsets that drive immune activation and neuronal damage during HIV infection in the central nervous system [128]. Using scRNA-seq on cerebrospinal fluid and blood from adults with and without HIV, Farhadian et al. identified a rare subset of myeloid cells associated with neurodegenerative-disease-associated microglia [128]. The authors were able to outline a potential mechanistic link between neuronal injury in HIV and the associated progression of neurodegenerative conditions. 

Kazer et al. performed scRNA-seq on PBMCs of four untreated individuals before, and longitudinally during, acute HIV infection to discover the gene responses that vary by time and cell subset in HIV infection [30]. They successfully identified gene expression responses that were concealed in bulk analysis: proinflammatory T cell differentiation, prolonged major histocompatibility complex II upregulation, and persistent natural killer (NK) cell cytolytic killing. The study provided a unified framework to study coordinated dynamic cellular responses in the early stages of HIV. This has significant implications in *Mtb*/HIV co-infection since LTBI reactivation is often associated with immune dysregulation in the very early phase of HIV co-infection.

Overall, single-cell technologies have contributed significantly to our understanding of cell subset diversity and individual cell heterogeneity in both TB and HIV. It has facilitated the identification of novel cell subsets and their molecular and functional characteristics during the infection process that can open the door for therapeutic and vaccine development strategies. 

## 4. Implications of scRNA-Seq in Animal Models of TB and HIV

The nonhuman primate (NHP) model remains the most accurate animal model that recapitulates the entire spectrum of TB as seen in humans [129,130]. Rhesus macaque (Macaca mulatta) is the most widely used model amongst all the NHP species. This is due to their social adaptability and genetic diversity, which makes them adept at investigating intra-specific variation [131]. The model is highly tractable and can be utilized to study latent TB and LTBI reactivation and can be co-infected with Simian Immunodeficiency Virus (SIV) [13,18,19,132]. Cataloging the diverse cellular architecture of the primate lung in LTBI and active TB is critical in understanding human disease progression. The multi-omics approach provides an atlas that serves as an open resource for identifying novel targets for disease interventions. scRNA-seq on dematricized single-cells from lungs of active-pulmonary-TB-infected rhesus macaques identified an influx of plasmacytoid dendritic cells (pDCs), the interferon (IFN)-responsive macrophage population, and activated T-cell responses [133]. On the other hand, LTBI lungs were characterized by a CD27+ natural killer cell subset. The study outlined the presence of unique immune populations associated with either TB control or dissemination, which could be targeted for therapeutics or vaccines in the future. Several studies have also attempted to generate an NHP cell atlas using scRNA-seq to provide a catalog of features that can be used to study human disease [134,135]. These studies have mapped the receptor and co-receptors for viruses causing human infectious diseases, then intersected the data with human genetic disease orthologues for translational associations. 

A novel approach combined bacterial fitness fluorescent reporter strains with scRNA-seq to dissect the functional heterogeneity of *Mtb*-infected alveolar and interstitial macrophages in a mouse model [136]. The approach showed that, while the local lung macrophage population was epigenetically constrained in response to the infection, an inter-species comparison confirmed the conserved nature of the macrophage subsets between mice and humans. Such a multimodal scRNA-seq approach has important implications for understanding the regulation of host cells before and during TB/HIV co-infection. Akter et al. employed scRNA-seq on *Mtb*-infected murine lungs to identify the enrichment of a type I Interferon (IFN) signature in the lymphocytes and a higher expression of Ly6A on the surface of activated lymphocytes [60]. The results are in concordance with the human TB signatures [137] and identify Ly6A as a critical marker of T-cell activation in TB. The dynamic changes in T-cell subsets during TB infection, measured using scRNA-seq, revealed a novel signature for exhausted CD4+ (*H1FX*, *ZFP36L2*, *VIM*, *PPP1R15A*) and CD8+ T cells (*ITM2C*) [138]. T-cell exhaustion has important implications in immune deficiency, exacerbation of acute TB, and CD8+ T cell function. The identification of different gene signatures and pathways associated with exhausted T-cell subsets can potentially facilitate a comprehensive understanding of the role of these subsets in *Mtb* pathogenesis in the setting of HIV co-infection. 

Similar to TB, the diverse and complex pathogenesis of HIV-1 remains a challenge in developing efficacious vaccines and cure strategies. The absence of treatment results in a highly variable disease progression with considerable heterogeneity in the phenotype of infected immune cells and their functions [139]. The heterogeneous nature of the HIV-infected cells also makes it difficult to target and treat the latent reservoirs [28]. The single-cell approach allows the study of the diverse mechanisms associated with HIV latency, including transcriptional repression and post-transcriptional blocks [140]. The effects of chronic HIV infection, including the persistence of viral reservoirs and neuropathogenesis, can be mimicked using Simian Immunodeficiency Virus (SIV) in an infected rhesus macaque model. The transcriptome of brain myeloid cells in cART-treated versus untreated macaques was examined using scRNA-seq. The results demonstrated that the virus induced changes in the brain microglia and that cART restored the homeostatic state of microglia like the uninfected control [141]. The findings provide a critical perspective on the role of microglia/macrophages as key targets of the SIV infection of the central nervous system. This finding has direct implications in elucidating the underlying mechanisms of alteration of the clinical signs and delays in the diagnosis of neurotuberculosis, the most devastating extrapulmonary form of TB in the setting of HIV infection. The differences in the neonatal versus adult responses to the HIV-1 envelope were demonstrated using a macaque model [142]. Single-cell transcriptomic analyses showed that neonatal macaques had a decreased immunosuppression transcriptome signature compared to adult macaques following HIV envelope immunization. Additionally, the neonates had a high transcriptome profile of *BCL2* in T cells and decreased *IL10RA* transcripts in T, B, and NK cells post-HIV envelope vaccination compared to adult macaques. These findings are especially critical to advance our understanding of the transcriptomic changes in neonates who are exposed to HIV and have TB. 

Though the NHP model of SIV has unequivocally illustrated the mechanisms of HIV pathogenesis, there are significant genetic and biological differences between lentiviruses and some HIV-specific interventions that cannot be tested in nonhuman hosts. Humanized mice represent an attractive alternative animal model that recapitulates key aspects of human HIV biology and pathogenesis [143,144]. Single-cell transcriptome analysis of human CD45+CD3-CD19- cells isolated from the spleens of humanized mice demonstrated a significant upregulation of the genes involved in type I IFN inflammatory pathways in the innate immune subsets [145]. In another study by Aso et al., the characterization of HIV-1-producing cells in a humanized mouse model using scRNA-seq demonstrated the heterogeneity of HIV-1-infected cells, including *CXCL13^high^* cells that provide clues for the development of an HIV-1 cure [146]. Humanized mice also support dual infections of *Mtb* and HIV in the lung, CNS, and other organs [147,148,149]. Co-infection in humanized mice reproduces several important aspects of microbial synergy, such as greater viral and bacterial burden and exacerbated inflammation. Applying single-cell RNA sequencing technology in this model would further enable the profiling of the complex virus, bacterial, and host dynamics during infection (Figure 3). It is a powerful tool that helps improve our understanding of the intercellular communication networks and the host–bacteria/virus interactions in immunologically relevant animal models.

## 5. Gaps in Knowledge and Future Directions

There are substantial gaps in our understanding of the TB-specific immune responses among PLHIV, including identifying treatment and vaccine targets within the TB-HIV care cascades. HIV-induced CD4+ T cell depletion is considered the primary cause of LTBI’s reactivation. However, recent work in the nonhuman primate model has shown chronic immune activation to be a leading cause of LTBI reactivation due to sustained inflammation and T-cell dysregulation, despite cART-induced viral suppression [18,19]. Using an in-depth multi-omics approach, it is feasible to identify the complex biological pathways perturbed by the co-infection of HIV with LTBI that remain impaired despite cART. Furthermore, a network-based analysis could determine the mechanisms of early co-infection events in an immunologically relevant animal model. The proteins and pathways identified could form the foundation for developing host-directed therapeutics and/or vaccines against TB/HIV co-infection. Integrating sc-RNAseq data with spatial transcriptomics could help identify the immune cell subsets and proteins impacted by treatment in the tissue. This will enable us to better understand the long-term efficacy of existing treatments and pave the way for precision medicine in TB/HIV treatment. Latently infected TB hosts, co-infected with HIV, are at risk of developing TB-associated immune reconstitution inflammatory syndrome (TB-IRIS) upon the initiation of cART. TB-IRIS is characterized by a rapid and exaggerated immune response resulting in worsening immunopathogenesis and mortality [150]. There is an urgent need to identify sensitive and specific biomarkers that can help predict TB-IRIS in these individuals and aid in developing treatment strategies to cure unmasked TB-IRIS. scRNA-seq also provides a tool to characterize the outliers that can help us understand drug resistance in high-risk populations. The CDC recommendation of a once weekly, oral dose of Isoniazid and Rifapentine (3HP) for 12 weeks of treatment of LTBI in high-risk areas of HIV has increased completion rates. However, a recent study in the NHP model of latent TB showed the persistence of bacilli in the lungs of two out of six animals, with evidence of ongoing inflammation upon treatment completion [132]. These results suggest that co-infected individuals may remain at risk for progression due to subsequent infections or reactivation due to persisting immune activation/inflammation. A multi-omics approach is needed to define biosignatures for treatment outcomes in TB/HIV co-infected cohorts.

Effective vaccines induce long-lasting protective immune memory, but the precise protection mechanisms are often poorly understood. Deploying sc-RNAseq to both the host and pathogen transcriptomes could identify and fast-track novel prophylactic candidates in TB/HIV co-infection. Sc-RNAseq could also be implemented on samples from human challenge studies to discover novel correlates of protection for different routes and regimens. Furthermore, the antigen-specificity of T cells in vaccinated individuals and cell trajectories can be predicted using scRNA-seq data. Pairing clonal expansion and transcriptome data could allow an unprecedented investigation into the heterogeneity of immune response in TB and HIV.

scRNA-seq is now a widely implemented tool in infectious diseases. However, improvements are needed in the ease of use of reagents, cost-effectiveness, and computational interpretation of the large datasets generated post-analysis. Additionally, there is a need to develop novel algorithms to ensure data reproducibility. Often, substantial efforts are needed to improve the methodology to achieve the targeted level of resolution for mapping cells of interest. Nevertheless, studying the transcriptomic landscape of thousands of single cells in TB- and HIV-infected complex multicellular organisms has facilitated important scientific advancements. Future studies integrating scRNA-seq profiling with functional phenotypes will help define the target modulation of pathways to achieve vaccine and therapeutic efficacy.

## Figures and Tables

**Figure 1 cells-12-02295-f001:**
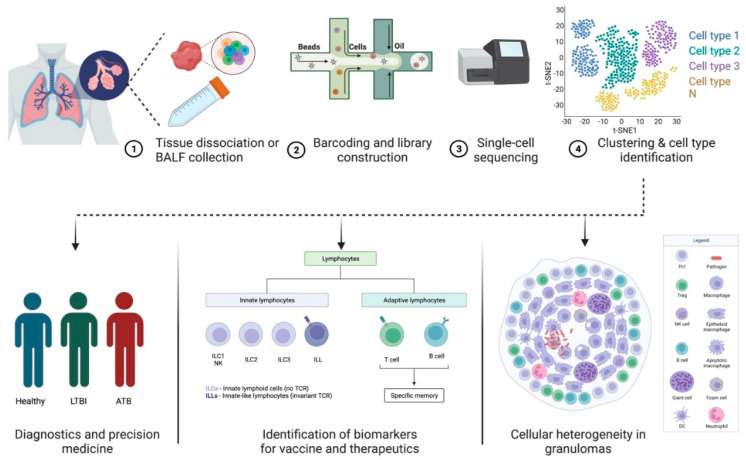
Applications of scRNA-seq in TB pathogenesis and diagnostics. (**1**) Lung or bronchoalveolar lavage fluid is collected from infected patients. (**2**) Single-cell suspension prepared from the sample is subjected to (**3**) barcoded library construction. (**4**) The cDNA libraries undergo next-generation sequencing that can be analyzed using different visualization tools. Data generated using scRNA-seq techniques can be used to study cellular heterogeneity in granulomas and identify biomarkers for vaccines and therapeutics in diagnostics and precision medicine.

**Figure 2 cells-12-02295-f002:**
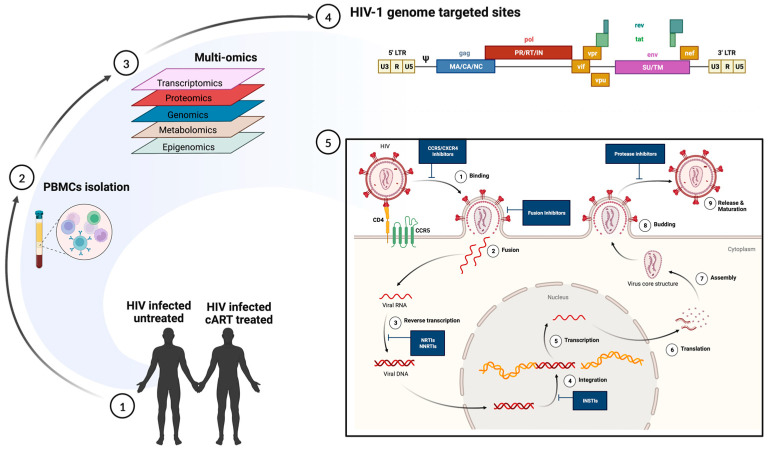
Multi-omics approach to understanding heterogeneity in HIV infection. (**1**) and (**2**) Polymorphonuclear cells, collected from HIV-infected untreated and HIV-infected cART-treated individuals, are subjected to multi-omics (**3**) including transcriptomics, proteomics, genomics, metabolomics, and epigenomics. The data generated can enable (**4**) an understanding of the transcriptionally active regions in productive infection and (**5**) help identify sites of therapeutic interventions.

**Figure 3 cells-12-02295-f003:**
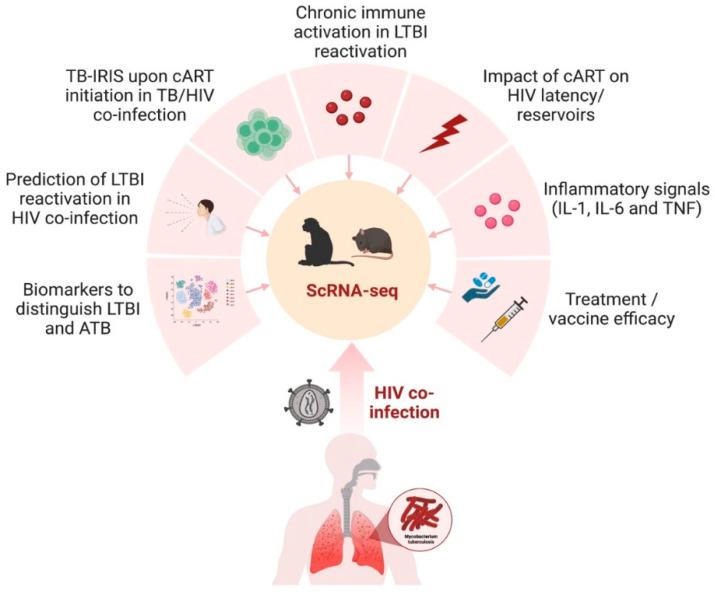
scRNA-seq in preclinical animal models to study TB/HIV co-infection. The nonhuman primate and humanized mouse model can be utilized to gain transcriptomic insights into several aspects of TB and HIV co-infection in humans, including the impact of treatment, diagnostics, inflammation, and immune activation.

## Data Availability

Not applicable.

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
