# Peer review of "Single-Cell Transcriptomics of Mtb/HIV Co-Infection"

_cells, 2023, doi:10.3390/cells12182295_

Round 1
Reviewer 1 Report
In their review, Kulkarni and co-authors describes single-cell transcriptomics of Mtb/HIV co-infection. his is a very well written and comprehensive review of the intra-cellular interactions in LTBI-HIV co-infection detected by RNA-sequencing
I only have few minor comments
Keywords: Suggest “RNA sequencing” to be included
Line 41: protection, not portection
Line 43: Virus instead of virus
Line 52: “…are more likely to die due to LTBI reactivation” – more likely than whom?
Line 119: “latent TB infection” has already been abbreviated as LTBI (line 40)
Line 150: The sentence “HIV continues to be a significant health
burden worldwide.” Is a repetition and should be deleted
Line 163: “intravenous drug users” is an obsolete term. Please use
“people who inject drugs” (PWID) instead
Line 199: Please explain abbreviations as “HAART”, GFP, FACS, RORC,
CDC, etc. the first time they are mentioned
Line 228: The abbreviation “LRAs” has been explained earlier
Line 375: “…human CD45+CD3-CD19-“ is something missing?
Line 512: “CDC recommended once weekly…” should be CDC´s recommendation of an once weekly….”,
In general, include space before the reference number
Author Response
In their review, Kulkarni and co-authors describes single-cell transcriptomics of Mtb/HIV co-infection. his is a very well written and comprehensive review of the intra-cellular interactions in LTBI-HIV co-infection detected by RNA-sequencing
I only have few minor comments
Keywords: Suggest “RNA sequencing” to be included
Line 41: protection, not protection
Response: The change has been made in Line 40.
Line 43: Virus instead of virus
Response: The change has been made in Line 51.
Line 52: “…are more likely to die due to LTBI reactivation” – more likely than whom?
Response: The sentence has now been modified to “Despite being on cART, people living with HIV (PLWH) have increased risk of LTBI reactivation” (Line 60-61).
Line 119: “latent TB infection” has already been abbreviated as LTBI (line 40)
Response: The change has been made (Line 158).
Line 150: The sentence “HIV continues to be a significant health
burden worldwide.” Is a repetition and should be deleted
Response: The sentence is now deleted.
Line 163: “intravenous drug users” is an obsolete term. Please use
“people who inject drugs” (PWID) instead
Response: The sentence is now deleted in response to Reviewer 2’s feedback.
Line 199: Please explain abbreviations as “HAART”, GFP, FACS, RORC,
CDC, etc. the first time they are mentioned
Response: The changes have been made (HAART – Line 338, GFP-Table 1, FACS – Line 209, RORC – Line 292 and CDC: Line 91).
Line 228: The abbreviation “LRAs” has been explained earlier
Response: The change has been made.
Line 375: “…human CD45+CD3-CD19-“ is something missing?
Response: The word cells has now been added (Line 439).
Line 512: “CDC recommended once weekly…” should be CDC´s recommendation of an once weekly….”,
Response: The change has been made (Line 495).
In general, include space before the reference number
Response: Space before reference number has now been added.
Reviewer 2 Report
This review, entitled "Single-cell transcriptomics of Mtb/HIV co-infection" aims at giving an overview on the current literature of single-cell RNA sequencing techniques for TB/HIV.
While the topic itself is interesting and highly relevant, with a lot of progress done in the past decade, this review fails - in my opinion - to give a good overview.
There major points of criticism are the following:
1) The review lacks a clear story-line. Of course, looking at a) HIV and scRNA seq; and b) TB and scRNA seq; there are tons of publications, with different points of focus.
However, this review should be about TB/HIV co-infection, but diverges a lot from this topic. For example, there are several paragraphs discussing the HIV latent reservoir (at different places in the review), which clearly would be enough for a separate review. There is no clear structure, no focus.
2) The actual topic "TB/HIV" is not discussed well - the whole "why we study TB/HIV combination". For example, the authors mention that it is hard to predict/test LTBI in HIV-infected patients. But why this is the case is not explained.
The authors should discuss, which TB tests are available, what is the current standard, and what makes it hard to test TB for HIV-infected patients (i.e. T cell dependent tests).
3) There are very outdated statements in the manuscript. One particular message is shockingly stigmatizing: lines 162-165: the authors state that not using intravenous drugs, or not being MSM "either completely protect or predispose" individuals to the risk of acquiring HIV.
This would mean heterosexual people not injecting drugs have a minor risk - well, have you looked at the statistics world-wide? Heterosexual women are actually affected most (globally)!!
This is a statement from 1987 - but until then, the epidemic has long reached the heterosexual population!!
4) Many well-known high-impact papers are not mentioned. The authors state (line 450-451): "While there is little to no published data on the use of scRNA-seq in the context of Mtb/HIV co-infection,...."
So two comments on this: A) if you say there is little to no published data: why do you aim to write a review article on exactly this topic? and B) There are papers out there - even if little, they should be mentioned and are most important here!
The review in the current form lacks a story-line, has misleading/outdated findings in there, and does not reflect the current state of the art.
Reviewer 3 Report
This a very important study Co-infection of TB and HIV continues to be a substantial health burden. HIV infection during TB predisposes the patient to LTBI reactivation. Single-cell RNAseg can identify new cellular heterogeneity
Author Response
This a very important study Co-infection of TB and HIV continues to be a substantial health burden. HIV infection during TB predisposes the patient to LTBI reactivation. Single-cell RNAseg can identify new cellular heterogeneity.
Response: The authors thank the reviewer for their positive feedback
Round 2
Reviewer 1 Report
The authors have revised accordingly